

# US adolescents' friendship networks and health risk behaviors: a systematic review of studies using social network analysis and Add Health data

Kwon Chan Jeon and Patricia Goodson

Department of Health & Kinesiology, Texas A&M University, College Station, TX, USA

## ABSTRACT

**Background.** Documented trends in health-related risk behaviors among US adolescents have remained high over time. Studies indicate relationships among mutual friends are a major influence on adolescents' risky behaviors. Social Network Analysis (SNA) can help understand friendship ties affecting individual adolescents' engagement in these behaviors. Moreover, a systematic literature review can synthesize findings from a range of studies using SNA, as well as assess these studies' methodological quality. Review findings also can help health educators and promoters develop more effective programs.

**Objective.** This review systematically examined studies of the influence of friendship networks on adolescents' risk behaviors, which utilized SNA and the Add Health data (a nationally representative sample).

**Methods.** We employed the Matrix Method to synthesize and evaluate 15 published studies that met our inclusion and exclusion criteria, retrieved from the Add Health website and 3 major databases (Medline, Eric, and PsycINFO). Moreover, we assigned each study a methodological quality score (MQS).

**Results.** In all studies, friendship networks among adolescents promoted their risky behaviors, including drinking alcohol, smoking, sexual intercourse, and marijuana use. The average MQS was 4.6, an indicator of methodological rigor (scale: 1–9).

**Conclusion.** Better understanding of risky behaviors influenced by friends can be useful for health educators and promoters, as programs targeting friendships might be more effective. Additionally, the overall MQ of these reviewed studies was good, as average scores fell above the scale's mid-point.

Corresponding author
Kwon Chan Jeon,
kcjeon@email.tamu.edu

## INTRODUCTION

The US Youth Risk Behavior Survey (YRBS) has captured trends in health-related risk behaviors among adolescents in grades 9 to 12 between 1991–2011 (*Centers for Disease Control and Prevention, 2012c*). In the report, the number of adolescents who "had sexual intercourse with four or more persons (15.3%)" (*Centers for Disease Control and Prevention, 2012d*), and "used chewing tobacco, snuff, or dip on school property on at least 1 day (7.7%)" demonstrated a general increase (*Centers for Disease Control and Prevention,*

*2012e*). On the other hand, the number of those who "ever had at least one drink of alcohol on at least 1 day" showed a slight decrease from 72.5% to 70.8% during that time (*Centers for Disease Control and Prevention, 2012a*).

Based on these data, we asked: "What may have shaped these trends over time?" In attempts to answer this question, researchers have indicated that risky health behaviors among adolescents are strongly influenced by their peers or friendship relationships (*Hall & Valente, 2007*; *Prinstein, Brechwald & Cohen, 2011*; *Rew & Horner, 2003*). Despite this knowledge, the literature on health risk behavior during adolescence has focused, traditionally, on individual adolescent risk taking behaviors (e.g., whether adolescents engaged in smoking, drinking or sexual activity, as well as frequency or intensity of engagement) as the unit of analysis. More recently, however, advanced analytical methodologies—including Social Network Analysis (SNA)—have led to the study of patterns in health risk behaviors influenced by peer or social contexts (e.g., friendship networks and affiliations).

Assessing patterns has highlighted the utility of SNA for in-depth understanding of the risky health behaviors among adolescents based on their relationships or interactions with other peers (*Ennett et al., 2006*; *Haas, Schaefer & Kornienko, 2010*). SNA is an optimal research tool because SNA maps out relationship networks among different people in a social group context (*Valente, Gallaher & Mouttapa, 2004*). Additionally, utilizing SNA, researchers can describe the patterns of structural connectivity using a visual analysis of the networks or by generating statistical descriptions (*Crnovrsanin et al., 2014*). Therefore, SNA can help understand various risk behaviors that can be affected by other people (*Smith & Christakis, 2008*) and can help researchers assess adolescents' risky behaviors within peer networks, as well as identify the structures of friendship ties that can influence behaviors.

In the US over the last decade, researchers have studied peer effects upon adolescents' health risk behaviors using network structure data from the National Longitudinal Study of Adolescent Health to Adult Health (Add Health). The Add Health study gathers data on adolescents' health risk behaviors from a stratified sample of high schools (grades 7–12) nationwide, thus generating representative data. Furthermore, the Add Health data focus on social contexts (i.e., friendships and family relationships) that influence adolescents' health-related behaviors (*Harris et al., 2009*). Data are collected from in-school questionnaires and in-home interviews with adolescents, their peers, parents, and school administrators (*Harris et al., 2009*).

Several researchers have analyzed the Add Health data using social network analysis, demonstrating that SNA is a useful method for assessing both the structure of peer relationships, and/or friendship networks (*Fujimoto & Valente, 2012a*; *Mundt, 2011*). These studies indicate that either friendship ties or peer effects among adolescents can function as causal factors directly influencing peers' risk behaviors such as drinking and smoking. Also, peer influences affect behaviors both positively and negatively, depending on adolescents' perceptions of friends' behaviors (*Sieving et al., 2006*). Despite its valuable contribution, research utilizing Add Health data varies in focus, with researchers examining many

different types of friendships and various adolescent behaviors, such as drinking, tobacco use, and sexual intercourse.

The purpose of this study, therefore, is to answer the following questions through systematically reviewing the extant literature: (1) Which risky health behaviors have been examined using SNA and the Add Health data; (2) What findings have been identified in this literature relevant to friendship networks' impact on adolescents' risk behaviors; and (3) What is the methodological quality of this body of literature?

Systematic literature reviews contribute to a body of literature by organizing and assessing scientific findings to effectively demonstrate both the accuracy and reliability of evidenced-based information (*Mullen & Ramirez, 2006*; *Mulrow, 1994*). A long-term goal of this review is to lend further validity to applying SNA as a method for studying adolescents' health-risk behaviors and assist future researchers in developing guidelines for implementing network-based intervention programs.

## BACKGROUND

National data, collected every two years by the YRBS and hosted by the Centers for Disease Control and Prevention (CDC), report that risky behaviors including tobacco use, drinking alcohol, and sexual intercourse at a young age have been health concerns for US adolescents for more than 20 years. For instance, between 18 and 47% of adolescents in grades 9 through 12 engaged in smoking , drinking alcohol, or were involved in sexual activity in 2011 (*Centers for Disease Control and Prevention, 2012c*).

These behaviors are the main health challenges for adolescents because continued risky behaviors are associated with increasing health problems. Previous studies have indicated that smoking and drinking alcohol at an early age can lead to poor health, an increased risk for alcoholism (*Englund et al., 2008*), and chronic diseases (e.g., cardiovascular and cancer) (*Sawyer et al., 2007*). Moreover, early sexual activity among adolescents can increase the risk of contracting sexually transmitted infections (STIs) (*Kaestle et al., 2005*) and the Human Immunodeficiency Virus (HIV) (*Parillo et al., 2001*).

In addition, researchers have reported that if adolescents are involved in a risk behavior, they are more likely to engage in different risk behaviors *simultaneously*. *Johnson et al. (2000)*, for instance, identified a correlation between tobacco use and alcohol consumption among adolescents. Authors found adolescents who smoke are more likely to engage in binge drinking, simultaneously. Likewise, adolescents used to drinking heavily are more likely also to smoke regularly.

### Sexual behavior

Adolescents who engage in unprotected sexual behaviors have a considerably higher risk of experiencing an unintended pregnancy or contracting STIs, including HIV (*Tapert et al., 2001*), than those who do not engage in these behaviors. As of 2011, the YRBS reported that the percentage of adolescents (grades 9 through 12) responding positively to the question "ever had sexual intercourse" was 47.4% (*Centers for Disease Control and Prevention, 2012d*). Although this percentage is high, it represents a decline: in 1991, more than half

(54.1%) of adolescents in grades 9 through 12 reported that they had engaged in sexual intercourse.

Due to its many negative health and psychosocial consequences (not the least of which are sexual abuse and statutory rape), having had sexual intercourse before the age of 13 is another problematic behavior among adolescents. Rates for this behavior have plateaued between 2001 and 2009 (*Centers for Disease Control and Prevention, 2012d*), and have dropped 4% (from 10.2% to 6%) compared to 1991.

## Alcohol use

In the United States, alcohol use by adolescents is illegal (under age 21) and also remains a public health problem because it is associated with different risk behaviors, including tobacco use and unprotected sexual intercourse. Data from the YRBS documents that, in 2011, an estimated 70.8% of adolescents reported they "ever had at least one drink of alcohol on at least 1 day" during their lifetime (*Centers for Disease Control and Prevention, 2012a*). This statistic shows the percentage fell 11% compared to 1991.

By 2011, 38.7% of adolescents reported that they had "had at least one drink of alcohol on at least 1 day" during the 30 days before the survey (*Centers for Disease Control and Prevention, 2012a*). This percentage dropped from 50.8% in 1991, a 12% decrease.

## Tobacco use

Smoking is related to morbidity and mortality, and is a leading cause of chronic diseases (e.g., cardiac disease and vascular disease). Although smoking under the age of 18 years is illegal in the US (*US Food and Drug Administration, 2014*), data from the YRBS in 2011 indicate that 44.7% of teens reported they "had tried cigarette smoking" (*Centers for Disease Control and Prevention, 2012e*). This rate has fallen by 25% since 1991 (70.1%). Another problematic smoking behavior among adolescents (i.e., "smoked cigarettes on at least 1 day") when assessed in 2011, indicated 18.1% were smokers (*Centers for Disease Control and Prevention, 2012e*). This rate has fallen 9.4% since 1991 (from 27.5%). While during the period 1991–1997 the rates had gradually increased to 36.8%, the numbers have steadily decreased during between1999 and 2011.

## Marijuana and cocaine use

For adolescents, marijuana and cocaine use can cause unexplained changes in personality or attitudes such as anxiety, poor social skills, interpersonal alienation, and poor impulse control. These drugs also can affect physical development (e.g., brain and nerve damage, respiratory problems, and blood pressure) (*National Institute on Drug Abuse, 2014a*; *Brook, Balka & Whiteman, 1999*; *Shedler & Block, 1990*; *Volkow et al., 2014*). Moreover, they can lead adolescents to other risky behaviors (e.g., sexual intercourse or drinking alcohol). Marijuana and cocaine use are illegal for adolescents in the US (*National Institute on Drug Abuse, 2014b*), yet in 2011, 39.9% of adolescents reported, in the YRBS, ever using marijuana "one or more times" (*Centers for Disease Control and Prevention, 2012b*). This rate has steadily increased by 8.6% since 1991. Between 1991 and 1999 the increase was even larger, from 31.3% to 47.2%.

During that same time-period, an estimated 6.8% of adolescents reported they "ever used any form of cocaine one or more times" (*Centers for Disease Control and Prevention, 2012b*). Moreover, from 1991 to 1999, the rates had slightly increased (5.9% to 9.5%).

## METHODS

To examine whether friendship networks on influence adolescents' risk behaviors in studies utilizing SNA and Add Health data, we adopted Garrard's Matrix Method to search the literature and qualitatively synthesize study findings (*Garrard, 2010*). We searched publications that specifically used the Add Health data, catalogued by the website for the National Longitudinal Study of Adolescent to Adult Health in Carolina Population Center at University of North Carolina at Chapel Hill. However, because the search engine in the website was limited, it became necessary to identify additional articles through other electronic bibliographies. We identified and retrieved, therefore, all peer-reviewed journal articles housed in three additional electronic databases (Medline, Eric, and PsycINFO), and searched using variations of MeSH terms combined with Boolean operators (e.g., sexual behavior, drinking behavior, adolescent, *and*, social network, *or* network analysis). Additionally, we searched reference lists from each study, for additional articles. Using the Scopus database, we conducted further searches based on the first and corresponding author(s)' names listed in the retrieved reports.

Searching databases for this review initially yielded 4,455 results. Of these, 2,147 were identified in the Add Health website, 2,240 in Medline, and 68 in Eric and PsycINFO. After identifying irrelevant topics and removing duplicates in an initial screening step, we identified 87 relevant studies. Among these, 73 were excluded based on our criteria. To be included in our review studies needed to: (1) be published in a peer-reviewed journal between 2003 and 2014; (2) be written in English; (3) use SNA to study friendship networks' influence on adolescents' risky health behaviors; (4) focus on adolescents (aged 12 to 18 years old) in grades 7 through 12 (as these are the grades utilized in the Add Health data); and (5) utilize the Add Health data. We excluded studies if (1) only abstracts were published; (2) articles did not use SNA to study adolescents' risky health behaviors; (3) studies employed SNA, but did not utilize the Add Health data; (4) studies focused on the relationship between friendships and adolescents risky behaviors; and (5) studies employed hypothetical models or simulation modeling to examine the Add Health data.

Thus, we identified 14 articles eligible for full-text review. Moreover, we retrieved 1 additional article through retrieved studies' reference lists, and through first and corresponding author searches in Scopus. This study was published in 2001, but we included it in this review, because it met our other criteria. Finally, 15 articles met our inclusion criteria, and became the final sample in this review (see Fig. 1 as an adapted PRISMA flow diagram) (*Moher et al., 2009*).

Subsequently, we employed a review matrix to organize the information extracted from each article. The review matrix (see Table 1) included information for each study on: authors, sample, focal variables (behaviors studied), purpose, use of theory, statistical analyses, key findings, and suggesting prevention/intervention programs.

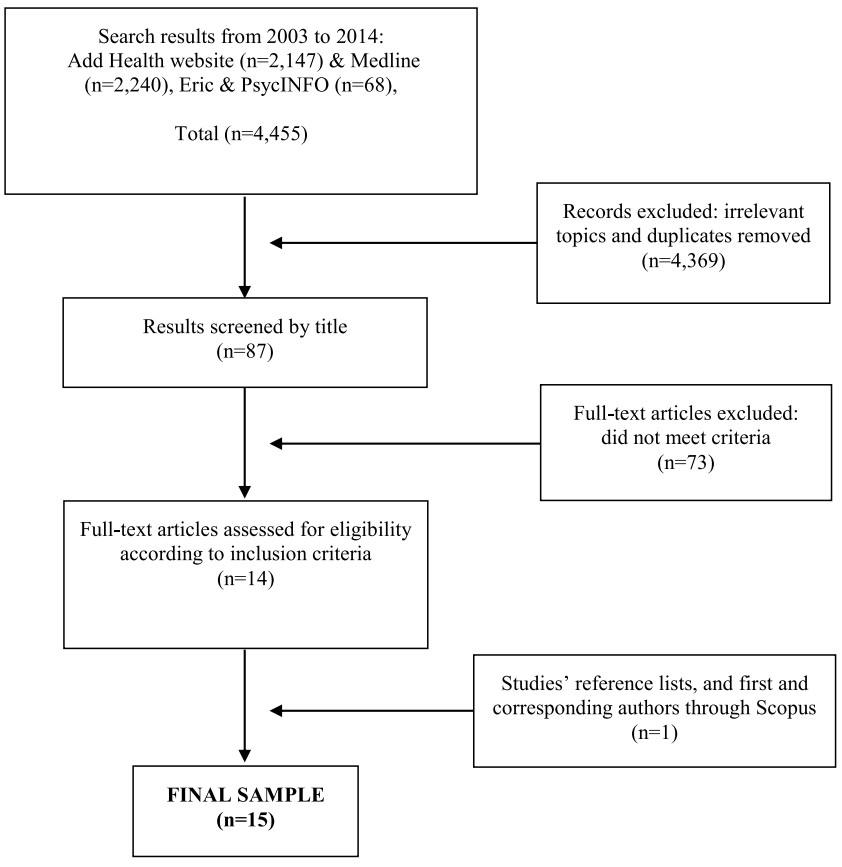

**Figure 1** **PRISMA flow diagram of reviewed studies.**

We assessed each article's methodological quality employing a numerical score that reflects specific features of a study's design and analyses (*Goodson, Buhi & Dunsmore, 2006*; *Jeon, Chen & Goodson, 2012*). In this review, each study received a methodological quality score (MQS), reflecting its performance on the criteria outlined in Table 2, to include: whether studies examined a single or multiple risk behaviors; if studies utilized an established theoretical framework; if the report contained visualizations of the networks; if the report presented visualizations of the analysis; if the study tested specific hypothesis; if the report explained the types of data analysis employed; and whether researchers made recommendations for developing programs, based on their findings. The scores ranged from 1 to 9 with a higher value representing better methodological quality.

## RESULTS

### Studies' characteristics

Fifteen studies met our inclusion criteria. Most studies ($n = 14$) were conducted in the US, and one paper, in France. Most reviewed studies ($n = 11$) were published between 2009 and 2014, perhaps because social network analysis only recently became popular as an analytical tool. Even though network data were collected in Wave I of the Add Health

**Table 1  Matrix of reviewed studies (by publication date).**

| Authors | Sample | Focal variables (behaviors studied) | Purpose | Use of theory[a] | Statistical analyses | Key findings | Suggesting prevention/ intervention program |
|---|---|---|---|---|---|---|---|
| Alexander et al., 2001 | 2,525 at Wave I | Cigarette smoking | "To investigate the effects of popularity, best friend smoking, and cigarette smoking within the peer networks on current smoking of seventh- through 12th grade students" | 3 | Logistic regression | "Having best friends who were cigarette smokers resulted in a twofold increased risk of current smoking (OR = 2.00)" "School smoking prevalence was positively associated with the odds of being a current cigarette smoker (OR = 1.73). For every 10% increase in school smoking prevalence, there was a 73% increase in the likelihood of current smoking" "There was a small but significant risk of being a current smoker for youth with higher levels of popularity and school smoking prevalence (OR = 1.08)" "The odds of current smoking were plotted against popularity for students with school smoking prevalence of 10%, a school with 25% smoking prevalence and one with a 40% smoking prevalence" | School policy |
| Jaccard, Blanton & Dodge, 2005 | 1,692 at Wave I & II | Sexual activity Binge drinking | "To gain a sense of the magnitude of influence that close friends may exert on adolescent health-risk behavior" | 1 | Logistic regression | "For sexual activity, of those individuals whose closest friend engaged in sexual activity across the two waves, 56% also engaged in sexual intercourse across the waves" "The unstandardized regression coefficient for the peer predictor at Wave 2 was 0.12 (95% CI [0.10–0.14], $p < .05$), suggesting that changes in the target's binge drinking behavior over time are associated with changes in the binge drinking behavior of his or her closest friend over time, holding constant friendship selection effects" "A statistically significant interaction effect was observed with the behavioral similarity between target and peer and peer binge drinking at Wave 2 (unstandardized regression coefficient for the product term = 0.15, 95% CI [0.06–0.25], $p < .006$)" | None |

Jeon and Goodson (2015), *PeerJ*, DOI 10.7717/peerj.1052

Table 1 (*continued*)

| Authors | Sample | Focal variables (behaviors studied) | Purpose | Use of theory[a] | Statistical analyses | Key findings | Suggesting prevention/ intervention program |
|---|---|---|---|---|---|---|---|
| *Sieving et al., 2006* | 2,436 at Wave I & II | Sexual intercourse | "To examine forms and pathways of friend influence on adolescents' sexual debut" | 1 | Logistic regression | "The odds ratio (1.01) suggests that for every 1% increase in sexually experienced friends at Wave 1, the odds that young people initiated sex by Wave 2 increased by 1%" "The more respect adolescents perceived they would gain from friends by having intercourse, the higher their odds of sexual intercourse (odds ratio, 1.2)" "...perceived respect from friends for having sex, the proposed mediator, was significantly associated with the proportion of sexually experienced friends ($r = .07; p = .015$) and with friends' attitudes about sex ($r = .14; p < .001$)" | Sex education programs, including "group norms for sexual behavior as well as the perceptions, skills and behaviors of individuals" |
| *Clark & Lohéac, 2007* | 20,745 at Wave I & II | Cigarettes/Marijuana Alcohol/Drunkenness | "To empirically evaluate the proposition that risky behavior by adolescents depends on the behavior of their peers (here, other adolescents in the same school)" | 3 | Regression | "If participation in drinking alcohol by the male peer group in the same school year increases by 25%, the adolescent's probability of drinking alcohol increases by 4.5%." "When the male peer group's alcohol participation in the same school year rises by 25%, the male's probability of drinking increases by 5.5%, with an analogous figure for females of 4.4%" "For cigarettes, an analogous rise in peer smoking increases the adolescent's probability of smoking by 2.2%..." | Policy |
| *Ali & Dwyer, 2009* | 20,745 at Wave I, II, & III | Smoking | "To empirically quantify the role of peer social networks in explaining smoking behavior among adolescents" | 2 | Multivariate structural model with fixed effects | "Having up to 25 percentage of close friends as smokers increases the probability of smoking by 5% (207/4), whereas being in a class containing up to 25% smokers increases the likelihood of smoking by 10%" | Public health interventions |

Jeon and Goodson (2015), *PeerJ*, DOI 10.7717/peerj.1052

Table 1 (*continued*)

| Authors | Sample | Focal variables (behaviors studied) | Purpose | Use of theory[a] | Statistical analyses | Key findings | Suggesting prevention/ intervention program |
|---|---|---|---|---|---|---|---|
| Pollard et al., 2010 | 6,696 at Wave I, II, & III | Tobacco use | "To examine how friendship networks in adolescence are linked to tobacco use trajectories through a combination of analytic techniques that traditionally are located in separate literatures: social network analysis and developmental trajectory analysis" | 2 | Latent class growth analysis | "Both perceiving that a greater number of one's best friends smoked, and increases in the perceived number of best friends who smoked over a one-year period, were associated with greater odds of an adolescent being in one of the smoking trajectories compared to being a never smoker" "Membership in a smoking group has these effects above and beyond the effect associated with the perceived number of best friends who smoke" | None |
| Lakon, Hipp & Timberlake, 2010 | 6,504 at Wave I | Smoking | "To examine adolescents' personal networks, school networks, and neighborhoods as a system through which emotional support and peer influence flow, and we sought to determine whether these flows affected past-month smoking at 2 time points, 1994–1995 and 1996" | 1 | Structural equation modeling | "…the popularity of adolescents (in-degree centrality) was affected both by their own past-month smoking and by their friends' smoking behavior. A 1% increase in past month smoking increased in-degree centrality by 2.3% ($b = 0.023: P < .01$)" | Using reciprocated friendships/popular youths to help stopping smoking Self-regulatory techniques (e.g., journaling) |
| Ali & Dwyer, 2010 | 20,745 at Wave I, II, & III | Alcohol consumption | "To empirically quantify the role of peer social networks in explaining drinking behavior among adolescents" | 2 | Multivariate structural model with fixed effects | "A 10% increase in close friends drinking will increase the likelihood of drinking by more than 2% (coefficient = 0.238, *p*-value = 0.000) and a 10% increase in drinking among grade-level peers is associated with a 4% increase in individual drinking (coefficient = 0.446, *p*-value = 0.000)" "An increase in drinking among individual's classmates by 10% will result in an increase in the likelihood of individual drinking and the frequency of alcohol consumption by approximately 4% (coefficient = 0.405, *p*-value = 0.005)" | Policy interventions at the school level |

Table 1 (*continued*)

| Authors | Sample | Focal variables (behaviors studied) | Purpose | Use of theory[a] | Statistical analyses | Key findings | Suggesting prevention/ intervention program |
|---|---|---|---|---|---|---|---|
| *Kreager & Haynie, 2011* | 898 at Wave I & II | Drinking | "To connect alcohol use, dating, and peers to understand the diffusion of drinking behaviors in school-based friendship networks" - "Test for the direct and indirect effects of partners and friends-of partners on individuals' problem drinking, net of individuals' prior drinking levels and the drinking of their immediate friends" | 1 | Hierarchical linear model | "Connections with drinking partners, friends, and partners' friends are all positively and significantly associated with future binge drinking. A standard deviation increase in (1) partner's prior drinking increases respondents' odds of binge drinking by 32 percent, (2) friends' prior drinking increases the odds of binge drinking by 30 percent, and (3) friends-of-partner prior drinking increases the odds of binge drinking by 81 percent" | None |
| *Ali & Dwyer, 2011* | 20,745 at Wave I | Sexual behavior | "To empirically quantify the role of peer social networks in influencing sexual behavior among adolescents" | 3 | Regression | "A 10% increase in close friends initiating sex will increase the likelihood of engaging in sexual intercourse by more than 2% and a 10% increase in sexual initiation among grade-level peers is associated with a 4% increase in individual sexual initiation" "Peer initiation of sex and the number of sexual partners of peers is statistically significant for the nominated peers and indicates that a 10% increase in sexual behaviors will result in a 4.7% increase in individual behavior" | Public health intervention |
| *Mundt, 2011* | 2,610 at Wave I & II | Alcohol use | "To investigate the association between adolescent social network characteristics identified in the previous studies, such as social status, social embeddedness, social proximity to alcohol users, and overall network interconnectedness, to adolescent alcohol initiation prospectively over time" | 3 | Generalized estimating equations | "Two of the 3 friend social network characteristics (ie, indegree, 3-step reach) increased the risk for the student to initiate alcohol use. For every additional friend with high indegree, the likelihood that an adolescent initiated alcohol use increased by 13% (95% CI, [4%–22%]). For every additional 10 friends within 3-step reach of a nominated friend, risk of alcohol initiation by a nondrinker increased by 3% (95% CI, [0.3%–6%]). Risk of alcohol use onset increased 34% (95% CI, [14%–58%]) for each additional friend who drank alcohol" | None |

Jeon and Goodson (2015), *PeerJ*, DOI 10.7717/peerj.1052

Table 1 (*continued*)

| Authors | Sample | Focal variables (behaviors studied) | Purpose | Use of theory[a] | Statistical analyses | Key findings | Suggesting prevention/ intervention program |
|---|---|---|---|---|---|---|---|
| *Fujimoto & Valente, 2012a*; *Fujimoto & Valente, 2012b* | 2,533 at Wave I | Drinking Smoking | "To identify some of the features or types of friendships that are most likely to affect adolescent alcohol use and cigarette smoking by computing the level of exposure to friends' behavior and their associations with individual behavior" | 3 | Logistic regression | "All friend adjusted odds ratios (AORs) were significant at $\alpha = .001$ level" "The effect from mutual friends (AOR = 2.07) on past-year drinking was slightly higher than exposures from outdegree-based unreciprocated alters (AOR = 2.02) or indegree-based unreciprocated alters (AOR = 1.97) on past-year drinking" "The effect of exposure from mutual friends on current smoking (AOR = 4.44) was almost 1.6 times higher than the effects of exposure from outdegree-based unreciprocated alters (AOR = 2.89) or indegree-based unreciprocated alters (AOR = 2.73) on current smoking" "The odds ratio for the mutual friendship (AOR = 4.44) falls above the upper 95% CIs for both outdegree-(upper 95% CI = 3.96) and indegree-based (upper 95% CI = 3.74) unreciprocated alters, which provides evidence that the differences in odd ratios were statistically significant" "The effect of ego-nominating friends (outdegreebased influence, AOR = 2.02) was a little bit higher than the effect of alter-nominating friends (indegree-based influence, AOR = 1.97) on past-year drinking, and similar results with regards to the effect of directionality of friendship on current smoking (AOR = 2.89 for outdegree-based influence and AOR 2.73 for indegreebased influence)" "The magnitude of the effect of outdegree-based influence from alters regardless of reciprocation on past-year drinking (AOR = 3.29) was much higher than the effect of influence from mutual friendship on past-year drinking (AOR = 2.07)" "The influence from the "best friends" was actually smaller than the combined influence of the remaining friends for past-year drinking (AOR = 1.55 for best-friends influence and AOR = 2.62 for the rest of the friends)" "Classmates' influence was significant for some types of friends' influence at $\alpha = 0.05$ level for drinking outcome" | School-based substance use prevention programs |

Jeon and Goodson (2015), *PeerJ*, DOI 10.7717/peerj.1052

Table 1 (*continued*)

| Authors | Sample | Focal variables (behaviors studied) | Purpose | Use of theory[a] | Statistical analyses | Key findings | Suggesting prevention/ intervention program |
|---|---|---|---|---|---|---|---|
| *Fujimoto & Valente, 2012b* | 12,551 at Wave I | Alcohol | "To investigate the relative strengths of two network influences on adolescent drinking (and drinking frequency), derived from affiliation with organized sports/club activities with their friends, using the affiliation exposure model" "To investigate how these different influence effects operate together as risk factors for adolescent drinking and drinking frequency, allowing us to disentangle overlapping influences from friend and nonfriend affiliates" | 2 | Ordinal logistic regression | "The affiliation influence through sports had a significant effect on both any drinking and frequent drinking (adjusted odds ratio AOR = 1.20; $p < .05$). This result indicates that greater alcohol exposure to sports member drinkers leads to a higher likelihood of any drinking (or frequently drinking)" "The influence through clubs had a significant effect on any drinking (AOR = 1.46; $p < .01$), but only a marginal effect on frequent drinking (AOR = 1.23; $p < .1$). These results indicate that adolescents exposed to drinkers in their sports or clubs were more likely to drink themselves, but the effect on frequent drinking was stronger in a sports context than in a club one" "The friends' exposure had a significant effect on both any drinking and frequent drinking (AOR = 1.55; $p < .001$), which indicates that adolescents with friends who drink were more likely to drink themselves" "The affiliation influence through sports members who were also friends had marginal effects on any drinking and frequent drinking (AOR = 1.08; $p < .1$), but the affiliation influence through club members who were also friends had a significant effect on any drinking and frequent drinking (AOR = 1.15; $p < .01$)" "The affiliation influence through nonfriend club members had a significant effect on both drinking behaviors (AOR = 1.37; $p < .01$)" "The effects of affiliation influence through fellow sports members who were also reciprocated friends became significant for both any drinking and frequent drinking (AOR = 1.16; $p < .01$)" "The magnitude of the effect through club members who were also reciprocated friends became larger and more significant (AOR = 1.22; $p < .001$) compared with the results of the nominated-friends' affiliation model (AOR = 1.15; $p < .01$)" "Affiliation influence through nonreciprocated friend club members was significant (AOR = 1.25; $p < .05$)" | School-based substance use prevention programs |

Jeon and Goodson (2015), *PeerJ*, DOI 10.7717/peerj.1052

Table 1 (*continued*)

| Authors | Sample | Focal variables (behaviors studied) | Purpose | Use of theory[a] | Statistical analyses | Key findings | Suggesting prevention/ intervention program |
|---|---|---|---|---|---|---|---|
| *Fujimoto & Valente, 2013* | 15,355 at Wave I | Drinking alcohol Smoking | "To investigate two contagion mechanisms of peer influence based on direct communication (cohesion) versus comparison through peers who occupy similar network positions (structural equivalence) in the context of adolescents' drinking alcohol and smoking" | 2 | Logistic regression | "The odds ratios for cohesion exposure to drinking were significant for all distances, with the highest in magnitude at distance one (OR = 1.57; $p < 0.001$), followed by distance two (OR = 1.44; $p < 0.001$), distance three (OR = 1.17; $p < 0.01$) and distance four (OR = 1.16; $p < 0.01$)" "The odds ratios for cohesion exposures to smoking were statistically significant up to distance two (but not significant for distances greater than two) with the highest in magnitudes at distance one (OR = 1.50; $p < 0.001$), followed by distance two (OR = 1.40; $p < 0.001$)" "The odds ratios for structural equivalence exposure to drinking were statistically significant for all distances, with the highest in magnitude at distance one (OR = 2.36; $p < 0.001$), followed by distance two (OR = 2.30; $p < 0.001$), distance three (OR = 1.90; $p < 0.001$) and distance four (OR = 1.88; $p < 0.001$)" "The odds rations for the structural equivalence exposure to smoking": "exposure effects were statistically significant for all distances with the highest in magnitude at distance one (OR = 1.99; $p < 0.001$), followed by distance two (OR = 1.83; $p < 0.001$), distance three (OR = 1.59; $p < 0.001$) and distance four (OR = 1.59; $p < 0.001$)" | School-based substance use prevention programs |

Jeon and Goodson (2015), *PeerJ*, DOI 10.7717/peerj.1052

Table 1 (*continued*)

| Authors | Sample | Focal variables (behaviors studied) | Purpose | Use of theory[a] | Statistical analyses | Key findings | Suggesting prevention/ intervention program |
|---|---|---|---|---|---|---|---|
| *Tucker et al., 2014* | 1,612 at Wave I & II | Marijuana use | "To examine whether three structural features of friendships moderate friends' influence on adolescent marijuana use: whether the friendship is reciprocated, the popularity of the nominated friend, and the popularity/status difference between the nominated friend and the adolescent" | 2 | Stochastic actor-based model (SAM) in R-Siena | "In school 1, there was a significant positive interaction between friends' influence on marijuana use and friend reciprocity (Table 3). Thus, adolescents tended to adopt the drug use behaviors of their mutual friends, whereas there was no evidence that they adopted the behaviors of friends who did not also nominate them as a friend" "The interaction between friend popularity and friends' influence on marijuana use was positive in both schools, but only statistically significant in school 2 (Table 4). In school 2, adolescents were likely to adopt the marijuana use behaviors of their more popular friends" | None |

**Notes.**

[a] 1 = "Reported a scientific/behavioral theory"; 2 ="Reported some theoretical explanation"; and 3 = "Reported no theoretical framework."

**Table 2** Methodological characteristics and frequency distribution of each criterion among 15 reviewed studies using social network analysis and Add Health Data.

| Methodological characteristic | Scoring options (maximum total score = 9 points) | Distribution of characteristics among 15 reviewed studies[a] | |
|---|---|---|---|
| | | Frequency (*n*) | Percent (%) |
| Number of behaviors | Focused on two or more behaviors = 2 points | 4 | 26.7 |
| | Focused on one behavior = 1 point | 11 | 73.3 |
| Theoretical framework | Reported a scientific/behavioral theory = 2 points | 4 | 26.7 |
| | Reported some theoretical explanation = 1 point | 6 | 40 |
| | Reported no theoretical framework = 0 point | 5 | 33.3 |
| Visualization of network | Provided visual graphs of network (in full or a sample) = 1 point | 1 | 6.7 |
| | Did not provide visual graphs of network = 0 point | 14 | 93.3 |
| Visualization of analysis | Provided visual graphs that help understand proposed analysis = 1 point | 4 | 26.7 |
| | Did not provide visual graphs that help understand proposed analysis = 0 point | 11 | 73.3 |
| Hypothesis testing | Tested a proposed hypothesis = 1 point | 7 | 46.7 |
| | Did not test a hypothesis = 0 point | 8 | 53.3 |
| Data analysis | Reported both descriptive and inferential statistics = 1 point | 14 | 93.3 |
| | Reported only inferential statistics = 0 point | 1 | 6.7 |
| Recommendations for developing programs | Makes recommendations for prevention/intervention programs = 1 point | 10 | 66.7 |
| | Makes no recommendations for developing programs = 0 point | 5 | 33.3 |
| Methodological Quality Score | Total possible maximum points = 9 | 4.6 (SD = 1.24) | |
| | | Actual range (2–7 points) | |

**Notes.**

[a] The frequency and percentages were calculated based on 15 reviewed studies.

survey (1994–1995), we found the earliest publication on social networks among the reviewed studies was published in 2001.

All reviewed studies appeared in journals with impact factors ranging from 1.638 to 4.266. Four were published in the *Journal of Adolescent Health*, and two studies were published in *Addictive Behaviors.* The other journals (the *American Sociological Review*, the *American Journal of Public Health, Health Psychology, Developmental Psychology, Social Science & Medicine, Academic Pediatrics,* the *Journal of Adolescence, the Journal of Health Economics,* and *Perspectives on Sexual and Reproductive Health)* published one report each.

## Studies' findings

1) *Which adolescents' risky health behaviors have been examined using SNA and the Add Health data?*

The studies in this review utilized SNA to examine adolescents' substance use— drinking, smoking, and marijuana use—and sexual behavior—specifically, sexual intercourse.

Eight studies examined adolescents' alcohol consumption behaviors, focusing on different aspects of adolescents' drinking (*Ali & Dwyer, 2010*; *Clark & Lohéac, 2007*;

*Fujimoto & Valente, 2012a*; *Fujimoto & Valente, 2012b*; *Fujimoto & Valente, 2013*; *Jaccard, Blanton & Dodge, 2005*; *Kreager & Haynie, 2011*; *Mundt, 2011*). Among these reports, six studied adolescents' *drinking frequency* as affected by best (or close) friends, peer group, affiliated members (e.g., sports and club activities), or direct (and indirect) friends (*Ali & Dwyer, 2010*; *Clark & Lohéac, 2007*; *Fujimoto & Valente, 2012a*; *Fujimoto & Valente, 2012b*; *Fujimoto & Valente, 2013*; *Mundt, 2011*). Moreover, two studies (*Jaccard, Blanton & Dodge, 2005*, and *Kreager & Haynie, 2011*) investigated adolescents' *level of drinking* (specifically, bingeing) as influenced by friends.

In the six studies focused on *drinking frequency*, researchers used various questions from the Add Health questionnaires, including: "During the past 12 months, on how many days did you drink alcohol?" and "Think of all the times you have had a drink during the past 12 months, how many drinks did you usually have each time?"; "Over the past 12 months, on how many days have you gotten drunk or 'very, very high' on alcohol?"; and "During the past 12 months, how often did you get drunk?" In the two studies examining *level of drinking* (specifically, bingeing), researchers used the following questions: "Over the past twelve months, on how many days did you drink five or more drinks in a row?" Both the adolescents and their friends were asked this question.

Seven studies focused on cigarette use or smoking behaviors among adolescents (*Alexander et al., 2001*; *Ali & Dwyer, 2009*; *Clark & Lohéac, 2007*; *Fujimoto & Valente, 2012a*; *Fujimoto & Valente, 2012b*; *Lakon, Hipp & Timberlake, 2010*; *Pollard et al., 2010*). All seven examined adolescents' *frequency of smoking* as influenced by various friendships, such as close or best friends, popular friends, mutual friends, or direct (and indirect) friends.

In these seven studies, authors used various questionnaire items, including: "During the past 30 days (past 12 months), on how many days did you smoke cigarettes?"; "During the past 30 days, on the days you smoked, how many cigarettes did you smoke each day?"; or "Of your 3 best friends, how many smoke at least 1 cigarette a day?"

Three studies investigated sexual behavior (intercourse) (*Ali & Dwyer, 2011*; *Jaccard, Blanton & Dodge, 2005*; *Sieving et al., 2006*). These studies examined the *frequency of sexual intercourse* as being influenced by close friends. Researchers used the questions: "Have you ever had sexual intercourse?"; "In what month and year did you have sexual intercourse most recently?"; or "If you had sexual intercourse, your friends would respect you more" from a section on "Motivations to Engage in Risky Behaviors."

Two studies (*Clark & Lohéac, 2007*; *Tucker et al., 2014*) focused on adolescents' marijuana use as influenced by peer groups. In this study, researchers used the questionnaire item: "During the past 30 days, how many times did you use marijuana?" In these studies, authors examined the *frequency of marijuana use* as influenced by adolescents' friendships.

Even though the Add Health questionnaires have items addressing two different behaviors in tandem (e.g., sexual intercourse + drinking; and sexual intercourse + drugs), none of the reviewed studies examined more than one behavior at a time.

(2) *What research findings have been identified in the literature relevant to friendship networks' impact on adolescents' risk behaviors?*

## Alcohol use

As mentioned previously, eight reviewed studies investigated the relationship between drinking alcohol and friendship networks (*drinking frequency and amount of drinking*). For instance, the study conducted by *Fujimoto & Valente (2012a)* examined how various friendship types influenced adolescents' substance use, including drinking (frequency). Authors classified three types of friendships: mutual friendships, directional friendships, and intimate friendships (see Fig. 2 for diagrams of the three types of friendships examined by *Fujimoto & Valente, 2012a*). A mutual friendship was defined as reciprocated friends (knowing each other as friends). A directional friendship was defined as an unreciprocated nomination that originated either from an ego or from an alter (i.e., ego-nominating friend and alter-nominating friend). An intimate friendship was defined as closest or best friends who were being first nominated (Fig. 2). These three friendship types were based on friendship nominations that students were asked to make as they nominated five best male friends and five best female friends from the Add Health data.

*Fujimoto & Valente (2012a)* found mutual friends were more likely to influence their friends' drinking behavior (frequency) than a directional friendship in the previous year (AOR = 2.07; $p < 0.001$). Moreover, in the directional friendships among unreciprocated alters, the authors found ego-nominating friends (Fig. 2) were slightly more influential in adolescents' drinking behavior than alter-nominating friends (AOR = 2.02; $p < 0.001$). Paradoxically, for the intimate relationships (Fig. 2), the study indicated that non-best friends were more likely to influence adolescents' past year drinking than best friends (AOR = 2.62; $p < 0.001$).

*Fujimoto & Valente (2013)* also examined the influence on adolescents' drinking (and drinking frequency) of friends and affiliated members in sports and club activities. Adolescents were asked in which school organized clubs or sports they participated. Based on this information, authors divided activities into 12 types of sports and clubs, such as playing chess, studying French, and basketball.

Moreover, *Fujimoto & Valente (2013)* categorized friendships as (1) all nominated friends (adolescent nominated the alter as a friend, the equivalent to "directional friendships" in Fig. 2), and (2) only reciprocated friends (both adolescents mutually called each other friends) (Fig. 2). *Fujimoto & Valente (2013)* then created affiliation models based on nominated friends (i.e., General affiliation—the influence from all members' friendships; Nominated-friends' affiliation—the influence from adolescents who were nominated as friends; and Nonfriends' affiliation—the influence from adolescents who were not nominated friends). Affiliation models based on reciprocated friends were also developed (i.e., General affiliation—the influence from all members' friendships; Reciprocated-friends' affiliation—the influence from adolescents who had at least one reciprocated friend; and Nonreciprocated-friends' affiliation—the influence of a nonreciprocated friend).

In the nominated friends' general affiliation model, sports members influenced adolescents' drinking and frequency of drinking (AOR = 1.20; $p < 0.05$), and club members only affected adolescents' drinking (AOR = 1.46; $p < 0.01$). This study additionally

A

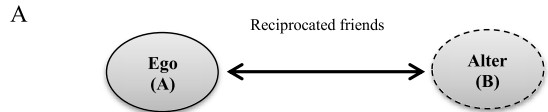

B

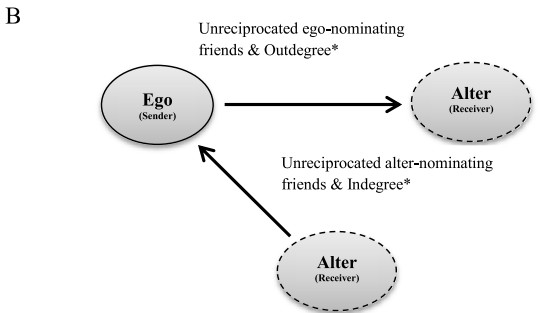

C

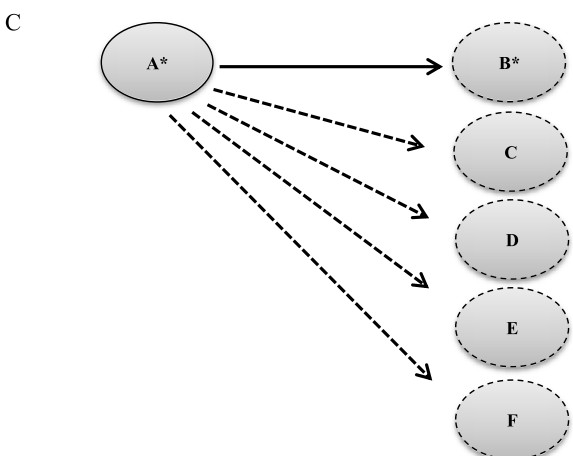

**Figure 2 Diagrams of the three types of friendships examined by *Fujimoto & Valente (2012a)*.** (A) Mutual/Reciprocated friendships. (B) Directional friendships: *Outdegree is the number of friendship ties that the ego who is a focal point within a network "*sends*" & *Indegree is the number of friendship ties that the ego "*receives*" (*Hall & Valente, 2007*). (C) Intimate friendships: *B was nominated as *best* or *close friends* by A; C–F were nominated as friends, but not *best* or *close friends.*

demonstrated that friends who drink were also more likely to affect adolescents' drinking and drinking frequency (AOR = 1.55; $p < 0.001$). In the nominated-friends' affiliation model, this study indicated club members significantly influenced adolescents' drinking and drinking frequency (AOR = 1.15; $p < 0.01$) (*Fujimoto & Valente, 2013*). In the nonfriends' affiliation model, club members who were not friends were more likely to affect drinking and drinking frequency of adolescents (AOR = 1.37; $p < 0.01$). In the reciprocated friends' affiliation model, sports members who were mutual friends with adolescents significantly influenced drinking and frequent drinking (AOR = 1.16; $p < 0.01$). Moreover, club members who were mutual friends were more influential in adolescents' drinking and frequency of drinking (AOR = 1.22; $p < 0.001$) than the results based on the nominated-friends affiliation model ($p < 0.01$). Additionally, in the nonreciprocated friends' affiliation model, club members significantly influenced drinking and drinking frequency of adolescents (AOR = 1.25; $p < 0.05$) (*Fujimoto & Valente, 2013*).

Uniquely, this study showed that club members who have no friendship ties with others influenced other adolescents' drinking behavior within the affiliation friendship networks. This finding can be explained by the fact that club members do not need to be intimate friends to be connected to each other, because they share many common interests and behaviors, even if they are not friends.

In another study, *Jaccard, Blanton & Dodge (2005)* evaluated how close friends influence adolescents' binge drinking. In this study, close friends were defined as those who were nominated by adolescents. Authors found a statistical significance in the behavioral similarity (binge drinking) between adolescents and their close friends ((unstandardized regression coefficient) 0.15; $p < 0.006$). Additionally, the study demonstrated that when adolescents' drinking behavior changed between Wave I (1995) and Wave II (1996) of data collection, their close friends' binge drinking also changed accordingly during the same time period ((unstandardized regression coefficient) 0.12; $p < 0.05$).

The other five studies showed similar findings, indicating that friendships that matter, among adolescents, were more likely to exert influence upon adolescents' drinking behavior. For instance, among these five studies, *Clark & Lohéac (2007)* found that "if participation in drinking alcohol by the male peer group in the same school year increases by 25%, the adolescent's probability of drinking alcohol increases 4.5%" (p. 773). Likewise, the study by *Ali & Dwyer (2010)* showed that if the number of close friends who drink increased by 10%, other adolescents' drinking would increase by 2%. Authors also found "a 10% increase in drinking among grade-level peers … associated with a 4% increase in individual drinking" (p. 340).

## Tobacco use

Seven of the fifteen studies reported friendship influence on adolescents' frequency of smoking. For instance, *Ali & Dwyer (2009)* categorized peer network as not only close friends who were nominated by the adolescents, but also those who were classmates and others from the same grade in school. A key finding from the study was that "having up to 25 percentage of close friends as smokers increases the probability of smoking by

5% ... whereas being in a class containing up to 25% smokers increases the likelihood of smoking by 10%" (p. 406).

In another study, *Fujimoto & Valente (2012b)* investigated the influence of peer networks on adolescent's substance use (smoking cigarettes and drinking alcohol), based on contagion mechanisms, in terms of cohesion and structural equivalence. Cohesion referred to relationships within a network, for which there are direct ties or exchange of influence. Structural equivalence referred to relationships among adolescents who occupy similar positions as others within friendship networks (see Fig. 3 for diagrams of cohesion and structural equivalence in a network). Authors defined peers as those who were nominated by friends. In their analysis, they utilized a network exposure model to assess both cohesion and structural equivalence measuring peers' risk taking in terms of social distances (at four steps away from other adolescents—friends of friends of friends of friends). The results indicated "the odds ratios for cohesion exposures to smoking were statistically significant up to distance two (but not significant for distances greater than two) with the highest magnitude at distance one (OR = 1.50; $p < 0.001$), followed by distance two (OR = 1.40; $p < 0.001$)" (p. 1957).

These findings suggest that direct or indirect friends (a friend or the friend of a friend) were more likely to influence adolescents' smoking behavior than friends at distance three or four (the friend(1)-of-a-friend(2)-of-a-friend(3), or the friend(1)-of-a-friend(2)-of-a-friend(3)-of-a-friend(4)). Moreover, the researchers found that for structural equivalence exposure to smoking, "... exposure effects were statistically significant for all distances with the highest in magnitude at distance one (OR = 1.99; $p < 0.001$), followed by distance two (OR = 1.83; $p < 0.001$), distance three (OR = 1.59; $p < 0.001$) and distance four (OR = 1.59; $p < 0.001$)" (*Fujimoto & Valente, 2012b*, p. 1957). These findings suggest that adolescents, who were one and two steps away in the network structure, were more likely to affect adolescents' smoking behavior than adolescents at three or four steps away in terms of social distances.

The other five studies showed similar results, namely, that various close friendships, such as best, popular, and mutual friends, were more likely to influence adolescents' smoking behavior than non-close friends. For instance, *Alexander et al. (2001)* indicated that if adolescents have best friends who are cigarette smokers, those adolescents' probability of smoking increases two fold. Similarly, the study conducted by *Pollard et al. (2010)* demonstrated that "... a greater number of one's best friends (who) smoked, and increases in the perceived number of best friends who smoked over a one-year period, were associated with greater odds of an adolescent being [a smoker]..." (p. 682).

### Sexual intercourse

Three studies focused on how adolescents' friendships influence each other's sexual behavior (specifically, intercourse). For instance, *Sieving et al. (2006)* classified close friends as those who were being nominated. Close friends were based on friendship nominations by students who were asked to nominate best male and female friends. Researchers found

A

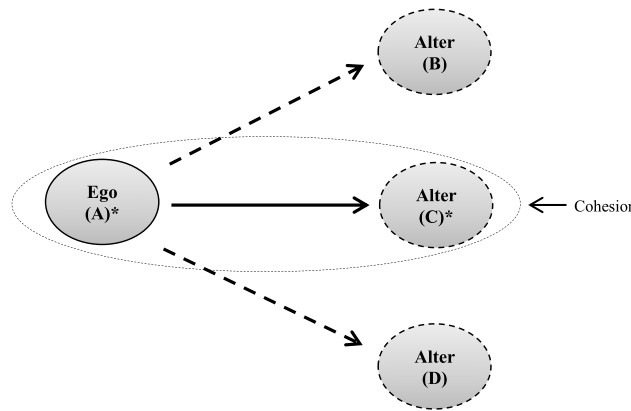

B

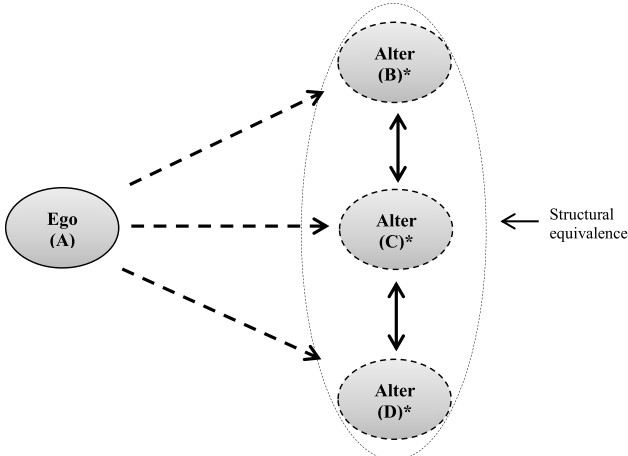

**Figure 3** **Diagrams of cohesion and structural equivalence in a network.** (A) Cohesion: *C has a direct tie with A and is influenced by A. The relationships between A–B and A–D are not cohesive, because the ties are indirect and there is no exchange of influence. (B) Structural equivalence: *B–C and C–D are structurally equivalent ties, because the individuals occupy the same position in the network.

that "…for every 1% increase in sexually experienced friends at Wave I (1995), the odds that young people initiated sex by Wave 2 (1996) increased by 1%" (p. 17).

In another study, *Ali & Dwyer (2011)* defined peer group as not only close friends who were nominated by adolescents, but also those who were classmates and others from the same grade in school. In this study, they found that if the number of close friends initiating sex increased by 10%, an adolescent's probability of initiating sex would also increase by 5%.

The third study (*Jaccard, Blanton & Dodge, 2005*), showed similar findings, indicating that close friends were more likely to exert influence on adolescents to engage in sex. They found "…of target individuals whose closest friends engaged in sexual activity across the

two waves (Waves I and II), 56% also engaged in sexual intercourse across the waves" (p. 141).

All reviewed studies examining sexual behaviors examined single behaviors; examined two or more behaviors in tandem, such as sexual activity with alcohol consumption, even though the Add Health data contains this information.

## Marijuana use

Among the 15 studies reviewed, two examined the influence of friendship networks on adolescent's marijuana use. *Tucker et al. (2014)* examined how different types of friendships (i.e., friend reciprocity, friend popularity, and popularity difference) affected adolescents' marijuana use in two saturated schools samples (i.e., school 1 and 2) at Wave I and II. Friend reciprocity was defined as mutual friends (being nominated by each other as friends). Friend popularity was defined based on "the total number of friendship nominations received by a nominated friend at a given wave…" (p. 68). Popularity difference was defined as "the difference in number of friend nominations received…" (p. 68). Researchers found that two friendships (i.e., reciprocity and friend popularity) were more likely to influence adolescents' marijuana use. Regarding friend reciprocity, authors found "in school 1, there was a significant positive interaction between friends' influence on marijuana use and friend reciprocity… adolescents tended to adopt the (marijuana) use behaviors of their mutual friends…" (p. 72). For friend popularity, the researchers found that there was only statistical significance in school 2, indicating popular friends were more likely to influence adolescents' marijuana use.

The second study, conducted by *Clark & Lohéac (2007)*, examined peer group effects on adolescents' alcohol, smoking and marijuana use. However, in this study, authors did not find that friendships influenced adolescents' marijuana use.

(3) *What is the methodological quality of this body of literature?*

As previously mentioned, we assigned a methodological quality score (MQS—with a possible range of 1 to 9 points) to each reviewed study. Table 2 presents the distribution of reviewed studies in terms of the MQS criteria. The average MQS was 4.6 (SD = 1.24), with actual scores ranging from 2 to 7 points. Among the 15 studies, seven (46.7%) scored below 4.5. The main problems affecting studies scoring below average were lack of visuals for the networks and/or for the analyses.

Eleven reviewed studies (73.3%) focused on studying a single behavior (most commonly, smoking or drinking). Four studies (26.7%) analyzed two or more behaviors, such as alcohol and tobacco use, but each behavior was examined separately. None examined two or more risk behaviors simultaneously (e.g., sexual intercourse with drug or alcohol consumption).

Regarding using or adopting theoretical frameworks, while the majority (10 studies) employed a theoretical framework, five studies (33.3%) failed to do so. Among the 10 studies employing a theoretical framework, six (40%) provided some theoretical explanation or rationale and four studies (26.7%) presented a scientific or behavioral theory: Social Learning Theory and Social Comparison Theory. Eight studies (53.3%)

did not test a hypothesis. Seven reports (46.7%) tested a proposed hypothesis such as "...influence from mutual friendships has stronger influence on adolescent drinking and smoking than non-mutual friendships" (*Fujimoto & Valente, 2012a*, p. 137) or "adolescents with higher proportions of sexually experienced close friends are more likely to initiate sexual intercourse than others" (*Sieving et al., 2006*, p. 14).

Only one reviewed study (6.7%) provided visual graphics for the networks examined, while four studies (26.7%) provided illustrations for how friendship influences egos and their alters. Fourteen studies (93.3%) employed and reported both descriptive and inferential statistics in their data analysis. One study (6.7%) reported only inferential statistics. More than half of the reviewed studies (66.7%) made recommendations for prevention or intervention programs, based on their network-related findings. *Lakon, Hipp & Timberlake (2010)*, for instance, propose in their study: "These friendship pairs could be targeted for a school-based intervention, either to help both adolescents in a pair remain nonsmokers or so that they could help each other stop smoking" (p. 1226–1227).

## DISCUSSION

This systematic review consolidated a segment of the current research employing SNA for studying adolescents' health risk behaviors. Specifically, we synthesized findings from network analyses based on the Add Health data, and assessed each analysis' methodological quality (presented in Table 2).

In this review, fifteen studies met our inclusion criteria. These studies found that, in general, various types of friendships exert influence upon adolescents' health risk behaviors. Across reviewed studies, having friends engaging in risky behaviors is a negative predictor of adolescents' healthy behaviors or a positive predictor of risky ones.

More than half of the reviewed studies examined friendship network effects on adolescents' risky behaviors either at a single point in time, or over time. Based on these studies, we learn that individuals who have friends or are linked to friendship networks exhibiting risky behaviors (e.g., smoking or alcohol consumption) are at increased risk for engaging in these behaviors both initially, and over time.

These findings from the Add Health data mirror results from a longitudinal study conducted in Finland. *Mercken et al. (2012)* assessed the relationship between substance use (alcohol consumption) and friendship networks among Finnish adolescents at different data points (i.e., time 1, time 2, time 3, and time 4). The results demonstrated that friends with risky drinking behaviors influenced adolescents to engage in similar drinking behaviors over time (between time 1 and 2). These results indicate, therefore, that SNA can account for the role of time in risky behaviors with more nuanced information than traditional longitudinal designs (*Rothenberg et al., 1998*).

The reviewed studies highlighted that SNA can help researchers better understand the complex mechanisms underlying the connection between friendships among adolescents and risky behaviors. Even studies that utilize SNA but are not included in this review mention SNA is a helpful tool for understanding adolescent behaviors as an outcome of social relationships, as well as for understanding changes in behaviors and friendship

networks over time (*Christakis & Fowler, 2007*; *Luke & Harris, 2007*; *Mercken et al., 2009*; *Smith & Christakis, 2008*), because friendship ties and behaviors occur inside the structure of dynamic interpersonal relationships among adolescents (*Steglich, Snijders & Pearson, 2010*). For instance, adolescents may choose friends having similar behaviors as theirs, or they may change their behaviors to develop new friendships or to match the behavior of existing friends. SNA, thus, can help explain peer selection, as well as lead to modeling changes in behaviors as a function of ties over time (*Christakis & Fowler, 2007*; *Luke & Harris, 2007*; *Scott & Carrington, 2011*).

In addition, SNA also allows better understanding of phenomena that cannot be adequately studied with traditional linear analyses. In particular, linear analysis cannot provide measures of structural linkages among individuals located inside a network, as a supplement to measures of an individuals' health risk behaviors. Using SNA, however, researchers are able to account for, and examine network dynamics and structure, such as density (i.e., the number of actual connections as a function of the total possible connections in a network) or degree (i.e., the number of ties, in and out, with other individuals in a network) (*Valente, 2010*), the impact of a network structure upon health behaviors, as well as the role of individuals as a function of their placement in the network. Moreover, SNA can create visualizations, depicting ties among individuals (*Scott & Carrington, 2011*; *Valente, 2010*), showing how an individual's position may act as a mediator for positive or negative behavioral influences. For instance, the reviewed study carried by *Kreager & Haynie (2011)* found that "indirect ties to a drinking peer through a romantic partner are associated with significantly higher future drinking than is the drinking of more proximal friends or romantic partners" (p. 756).

When SNA is employed in the study of health behaviors, it can not only identify structural and relational factors associated with behavioral changes in individuals or groups, but also provide information that can be used for developing effective network-based intervention programs to reduce health risk behaviors. In a study conducted by *Valente et al. (2007)*, for instance, the authors compared changes in adolescents' substance use (i.e., cigarette, alcohol, marijuana, and cocaine) between a control group receiving an evidenced-based prevention program and a network group receiving peer-leader intervention as a network prevention program. The results indicated that using a peer-leader program targeting the network was more effective in reducing substance use after a one-year follow-up assessment.

When assessed for overall methodological quality, the mean MQS for the studies reviewed herein was 4.6, an indicator of good quality relative to our seven criteria (a theoretical range of 1–9 points). Although the body of evidence we reviewed exhibits good methodological quality, because scores fell above the theoretical mid-point of our scale, not supplying illustrative visualizations showing the connections among individuals in networks, the absence of theoretical frameworks, and not examining two or more behaviors in tandem, affected the overall quality of this body of research, vis-à-vis our criteria.

One common weakness was the absence of either graphs depicting the networks or visualizations that could help understand the proposed analyses. Providing visualizations can

improve the clarity of, and highlight structural relationships within networks, more easily (*Crnovrsanin et al., 2014*). For instance, *Mundt (2011)* depicted a visual network of alcohol initiators and alcohol abstainers from their sample. The graphics the author provided help us understand not only the relationship among these adolescents, but guide us to a better understanding of the network measures themselves (i.e., differences between networks' density and degree are made more meaningful, when accompanied by a visual aid).

Another methodological weakness we identified was the reviewed studies' lack of a theoretical framework to examine adolescents' risky behaviors. The absence of a theoretical framework in research can lead to overlooking of salient factors and examining spurious ones. Conversely, using a theoretical framework can facilitate identifying possible causes (*Goodson, 2010*; *Green, 2000*). Theory helps develop programs, and findings from studies that use theory can be useful for determining the type of intervention that best suits risky behaviors. Understanding of social networks is growing, based on the increasing amounts of data being collected. Nonetheless, in order to develop effective interventions that target adolescents' networks, theoretical explanations of the mechanisms affecting behaviors within a network become even more important. Using available theories of networks, adolescent development, and structural influences on behavior, researchers can shed light upon the data they are now collecting and, over time, build the knowledge-base on this topic.

A further weakness of the reviewed studies was examining behaviors, individually. Studies focusing on two or more behaviors in tandem would allow for a better holistic understanding of the role of friendship networks in the dynamics of adolescents' risky behaviors, given that risk behaviors rarely happen in isolation. There is abundant evidence documenting adolescents' engagement in multiple risky behaviors carried out simultaneously. In the review conducted by *Cooper (2002)*, for instance, the author found that college students who drink alcohol were also involved in having sexual intercourse. Similar to *Cooper*'s review (*2002*), a study from *Johnson et al. (2000)* also identified that when teens engage in high levels of alcohol consumption, they also were more likely to smoke.

Similar to the reviewed studies, this review also carries important limitations. First, despite our attempt to locate all studies employing SNA utilizing the Add Health data, it is possible our search did not capture all existing studies, given that we limited the search to published reports. Second, to assess the methodological quality of this literature, we adopted and created the MQS criteria based on previous systematic reviews. The precise criteria we use in this review, therefore, have not been tested for their ability to generate valid and reliable assessments and could, therefore, be biased.

Despite these limitations, this systematic review demonstrated the important role of friends and friendship networks on adolescents' risky behaviors and the benefit of a SNA approach for better understanding of this role and its complex mechanisms. Moreover, this review is unique not only because it lends further validity to SNA as a method, but also because it synthesizes findings from high quality studies based on a national sample. Identifying how friendships or friendship networks function as pathways for adopting

risky behaviors can also help health educators and promoters to design guidelines for network intervention programs to reduce adolescents' risky behaviors.

### Funding

The authors declare there was no funding for this work.

### Competing Interests

The authors declare there are no competing interests.

### Author Contributions

- Kwon Chan Jeon conceived and designed the experiments, performed the experiments, analyzed the data, contributed reagents/materials/analysis tools, wrote the paper, prepared figures and/or tables.
- Patricia Goodson conceived and designed the experiments, reviewed drafts of the paper.

### Supplemental Information

Supplemental information for this article can be found online at http://dx.doi.org/10.7717/peerj.1052#supplemental-information.

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
