# Peer review of "US adolescents’ friendship networks and health risk behaviors: a systematic review of studies using social network analysis and Add Health data"

_PeerJ, doi:10.7717/peerj.1052_

## Round 0.1 · original submission · Major Revisions

Thank you for your submission. The reviewers felt that the paper was of interest, but that there are several areas that require further work' particularly with reference to the validity of your findings. Please note the attached comments and amend accordingly.

·

Basic reporting

Some elements should be more detailed.

Line 36: Explain 'individual adolescent risk taking behaviours' in greater depth
Line 43: Describe SNA more. Perhaps a diagram would help illustate the network mapping
Line 59: Do you mean 'either' or 'both'? This needs clarifying
Line 101: Provide a reference
Line 106: Explain why this is problematic
Line 111: Is under-21 considered adolescent?

Experimental design

Line 177: 86-73 is 13, but you say you had 14 papers (error carried forward to PRISMA diagram)
The number of decimal places used when reproting p-values needs to be consistent
A table showing the samples, research question and brief findsing for each paper might be useful

Validity of the findings

Line 472: I would avoid use of the word 'claim'

Additional comments

The paper is a somewhat verbose and could do with some tightening up. It is an interesting paper, but would read better if it were a little more concise.

Reviewer 2 ·

Basic reporting

The aim of the manuscript to systematically review the existing research on adolescent risky health behaviours using Social Network Analysis (SNA) and the ADD Health data. The authors did a systematic and thorough job on identifying the relevant studies. I feel that the introduction can be improved by better organisation. For example, the background subheading can be removed and merge the sections into Introduction. Throughout the introduction I feel that too much quotations were used (throughout the manuscritp) and the text should be paraphrased (except the items in the ADD Health that were used). The sections under Background, rather than just summarising the prevalent rate of each risky behaviour, previous findings of the effect of friendship or peer network/group should be added to each behaviour as well. This would set up for discussions about how using SNA to examine the effect of peer network would advance new knowledge in this field. In addition, in these sections some of the rates quoted do not match the ones presented in Figure 1 so Figure 1 seems irrelevant here.

Experimental design

The identification of studies was done in a systematic way. There is a couple of issues regarding the Method. What is the reason for limited the publication dates from 2003 onwards? One article was identified from 2001 and later in the manuscript it was noted that SNA is a relatvely new technique anyway so limiting the dates seems to be unnecessary. At Ln188, rather than referring the methodological quality criteria to previous review, it would be clearer to readers to have a proper description and background of the MQS here.

Validity of the findings

At Ln 207, it is not clear what the inclusion of the impact factors and the list of journals here would be relevant to the study goal. The presentation of findings under the 2nd research question seems repetitive at times and it was not clear which study the authors were referring to. For example, at Ln 284, it is not clear which "first study" the authors was referring to; "researchers" and "authors" were used in the next several sections which are findings of the one study. The names of the study author should be used instead to avoid confusion. Findings from Fujimoto & Valante was presented in almost 2 pages but findings from the other 7 studies were only presented in 2 paragraphs. What is the significant of this particular study? And there was no further discussion later on in Discussion. At Ln 289, descriptions of the friendship dyads are repeated here and it seems to retardant. Other results of significance presented for other behaviours looks like all only significant results were included? What about insignificant results? In addition to p-values, coefficients should also be included. Moreover, it is not clear whether Figures 3 and 4 were created based on the review findings or just generic diagrams to aid understanding of the different types of dyads.

The Discussion was not written in a way to highlight the unique findings of the review. The authors listed some advantages using SNA to study peer network (from Ln 469 to 504) but these information was unique from this review. What data SNA can analysed and what research questions can be examined is already known. Also I'm not sure if a cut-off of 4.5 of the MQS was appropriate; given that the average score was only slightly above 4.5 there is a need to discuss any studies that have a score of 4.5 and below.

It is confusing to read sections on discussing weakness of the reviewed studies and the limitations on the current review right after. Given the matrix of the reviewed studies included key findings, I'm not sure why it was only included in the appendix and not in the manuscript? Relating to this, it was mentioned that some studies did based on theories / theoretical frameworks in the text but under the "theory" column in the table it was listed "none" against all studies so there seems to be a mismatch here. Also under the "key findings" column paraphrasing should be used and not direct quotations from the reviewed studies.

---

## Round 0.2 · accepted · Accept

Thank your for your resubmission. I note that you have addressed the concerns of the reviewers, and have made adequate justification where you have chosen not to make suggested changes. I am pleased to accept your article for publication.